# Prevalence, socio-demographic and environmental determinants of asthma in 4621 Ghanaian adults: Evidence from Wave 2 of the World Health Organization's study on global AGEing and adult health

**Justice Moses K. Aheto**[1]*, **Emilia A. Udofia**[2], **Eugene Kallson**[3], **George Mensah**[2], **Minicuci Nadia**[4], **Naidoo Nirmala**[5], **Somnath Chatterji**[5], **Paul Kowal**[5,6], **Richard Biritwum**[2], **Alfred E. Yawson**[1,2]

1 Department of Biostatistics, School of Public Health, College of Health Sciences, University of Ghana, Accra, Ghana, 2 Department of Community Health, School of Public Health, College of Health Sciences, University of Ghana, Accra, Ghana, 3 Deloitte Consulting, West Africa Deloitte & Touche, Accra, Ghana, 4 National Research Council, Institute of Neuroscience, Padova, Italy, 5 World Health Organization HIS/HIS/MCS, Geneva, Switzerland, 6 University of Newcastle Research Centre for Gender, Health and Ageing, Newcastle, Australia

* justiceaheto@yahoo.com, jmkaheto@ug.edu.gh

**Data Availability Statement:** Data is freely available upon making official request to WHO

## Abstract

### Background

A previous multi-site study involving lower- and middle-income countries demonstrated that asthma in older adults is associated with long-term exposure to particulate matter, male gender and smoking. However, variations may occur within individual countries, which are relevant to inform health promoting policies as populations live longer. The present study estimates asthma prevalence and examines the sociodemographic characteristics and environmental determinants associated with asthma in older adults in Ghana.

### Methods

This study utilised data from the nationally representative World Health Organization Study on global AGEing and adult health (SAGE) Ghana Wave 2. A final sample of 4621 individuals residing in 3970 households was used in analytical modelling. Factors associated with asthma were investigated using single level and multilevel binary logistic regression models.

### Results

Asthma was reported by 102 (2.2%) respondents. Factors associated with asthma in the univariate model were: those aged 60–69 (OR = 5.22, 95% CI: 1.24, 21.95) and 70 or more (OR = 5.56, 95% CI: 1.33, 23.26) years, Ga-Adangbe dialect group (OR = 1.65, 95% CI: 1.01, 2.71), no religion (OR = 3.59, 95% CI: 1.77, 7.28), having moderate (OR = 1.76, 95% CI: 1.13, 2.75) and bad/very bad (OR = 2.75, 95% CI: 1.58, 4.80) health state, and severe/extreme difficulty with self-care (OR = 3.49, 95% CI: 1.23, 9.88) and non-flush toilet facility

SAGE Team through the WHO website at http://www.who.int/healthinfo/sage/cohorts/en/. The data is provided to researchers freely but the WHO SAGE Team only releases the data to researchers upon request made directly to WHO, and individual researchers granted permission to use the data are not allowed to make the data (in any form) available to third parties. Any third party interested in the data must apply directly to the WHO SAGE Team.

**Funding:** Funding was obtained for the main WHO SAGE Wave 2 Survey in Ghana with financial support provided by the US National Institute on Aging through Interagency Agreements (OGHA 04034785; YA1323-08-CN-0020; Y1-AG-1005-01) with the World Health Organization and a Research Project Grant (R01 AG034479- 64401A1). The funder provided support in the form of salaries for authors [RB, AEY], but did not have any additional role in the study design, data collection and analysis, decision to publish, or preparation of the manuscript. The specific roles of these authors are articulated in the 'author contributions' section. However, the work in this study did not receive any funding support. Also, Deloitte Consulting, West Africa Deloitte & Touche did not provide any financial support for the study and played no role in the study".

**Competing interests:** All authors except Mr Kallson declare that they have no conflict of interest. Though Mr Kallson was working with Deloitte Consulting, West Africa Deloitte & Touche at the time of the study, Deloitte Consulting did not provide any financial support for the study and did not play any role in the study design, data collection and analysis, preparation of the manuscript, or decision to publish. Mr Kallson participated in this study in his capacity as an individual and was not acting on behalf of Deloitte Consulting, and this does not alter our adherence to PLOS ONE policies on sharing data and materials.

(OR = 0.62, 95% CI: 0.39, 0.99). Factors independently associated with asthma in the adjusted models were: those aged 60–69 (OR = 4.49, 95% CI: 1.03, 19.55) years, father with primary education or less (OR = 0.40, 95% CI: 0.17, 0.94), no religion (OR = 2.52, 95% CI: 1.18, 5.41), and households with non-flush toilet facility (OR = 0.58, 95% CI: 0.35, 0.96). Significant residual household-level variation in asthma was observed. Over 40% of variance in asthma episodes could be attributable to residual household-level variations.

## Conclusion

Individual as well as household factors were seen to influence the prevalence of asthma in this national survey. Clinical management of these patients in health facilities should consider household factors in addition to individual level factors.

## Introduction

Over 544 million people worldwide had a chronic respiratory disease (CRD) in the year 2017, representing a rise of 39·8% compared with 1990, with asthma (3.6%) remaining the second most prevalent CRD after chronic obstructive pulmonary disease (3.9%). Cumulatively, there was a slight reduction in asthma prevalence from 1990 (3.9%) to 2017 (3.6%) [1]. In 2015, over 3 million people worldwide died from asthma and chronic obstructive pulmonary disease, with an increase in prevalence of 12.6% between 1990 and 2015 particularly in lower income countries [2]. Asthma remains a concern even into older ages, where chronic obstructive pulmonary disease comorbidity is more often overlaid [3]. Asthma is a chronic inflammatory disorder of the airways characterized by bronchoconstriction and mucus plugs that limit airflow [4]. Asthma development is linked to complex interactions between genetic and environmental factors [5].

The Global Initiative for Asthma estimates that asthma affects 300 million people globally and 50 million people in Africa [6]. Asthma prevalence varies across Africa, as reported in several independent, cross-sectional studies providing country estimates of 2.7% in Cameroon (≥19 years) [7], 4.3% in Gambia (≥15 years) [8], 5.3% in Botswana (10–64 years) [9], 6.5% in Tunisia (2–52 years) [10], 6.8% in Uganda (≥35 years) [11], 6.9% in Kinshasa (≥18 years) [12] and 15.2% in Nigeria (18–65 years) [13]. The range of ages included in specific studies, along with differences in case definition, make comparisons difficult across countries in and outside of Africa. The differences observed within and between countries can be attributed to differences in ascertainment of asthma (symptom-based questionnaires and/or physician diagnosed), susceptible populations (genetic/host factors) and environmental exposures. The World Health Organization (WHO) estimates an annual national incidence rate of 1.5/1000 [5], compared to 2.8/1000 in Tunisia and 4.6/1000 in Algeria [14]. Asthma care is focused on prevention and management of asthmatic attacks, in addition to the use of medication to alleviate or control symptoms [15]. Both old age and poor asthma control are associated with impaired quality of life. A previous study using nationally representative data in Ghana, indicated that ageing was associated with a lower quality of life. The study indicated that relative younger age positively influenced subjective well-being among the older adult population [16].

Global literature indicates that risk factors for asthma include female sex [17], a family history [18], maternal smoking during pregnancy [19], obesity [20], family size [18, 21, 22], age [23], sex [23], reduced anti-oxidant intake [24, 25], urban residence [26, 27], reduced exposure to childhood infections [5] and consumption of fast foods, especially hamburgers [28].

Environmental exposures which might trigger attacks include bacterial endotoxins, particulate matter, ozone, cockroaches, and house dust mites [5, 17]. A study in South Africa demonstrated additional exposures in older adults including living within two kilometres of mine dumps and the use of paraffin as a cooking/heating fuel [29]. Jie and colleagues describe the most common indoor factors associated with asthma as mould growth and environmental tobacco smoke and fuel combustion, particularly biomass fuels [27]. Such indoor conditions may be facilitated under conditions where heating and ventilation are poor [30]. They also reported a higher prevalence in urban compared to rural adults and differences in prevalence correlated with exposure to house dust mites, higher levels of vehicle emissions and a westernized lifestyle among urban dwelling adults [27].

Grass mats, animal dander, obesity, helminthiasis, pesticides and female sex have also been reported as contributing factors in studies conducted in African countries [17]. A study conducted in China revealed that women, age, smoking, having a first degree relative with asthma or pollinosis, combined with allergic conditions such as eczema, allergic rhinitis and gastro-esophageal reflux disease were associated with asthma [31]. Occupational exposure has been reported among gold miners (47.55%) in a study conducted in Obuasi, Ghana [32]. Nurses, poultry workers, hairdressers and wood cutters are other occupational groups at risk [17, 22].

Most studies providing evidence for risk factors in Ghana have been conducted among children [18, 33], while studies in older adults remain relatively limited. To address this gap, the present study examines socio-demographic and environmental determinants associated with asthma in older adults using data for Ghana from World Health Organization's Study on global AGEing and adult health (SAGE).

## Materials and methods

### Data source and sampling

SAGE is a multi-country longitudinal study that collects data to complement existing ageing data sources to inform policy and programmes. Data from the SAGE Ghana Wave 2 nationally representative sample which is a household-based survey was used for this study. The study employed multistage cluster sampling strategies where clusters were systematically sampled with known non-zero selection probability and households residing in the selected clusters identified/listed and individuals in those selected households selected for interview. WHO and the University of Ghana Medical School Department of Community Health collaborated to implement SAGE Wave 2 in 2014–2015. Detailed description of the methods is published elsewhere [34].

### Study population

Persons aged 50 years and above as well as a smaller sample of those aged 18–49 years were interviewed regarding their chronic health conditions and health services coverage, subjective wellbeing and quality of life, health care utilization, perceived health status, risk factors and preventive health behaviours, socio-demographic and work history, social cohesion and household characteristics. Further details about SAGE can be found through the WHO website (http://www.who.int/healthinfo/sage/cohorts/en/) including detailed information about SAGE Ghana Wave 2. In households identified as "older" for sampling purposes, all household members aged 50 years and older were invited to participate in the study.

### Outcome variable

The primary outcome of interest was asthma status based on self-report based on the question, "Have you ever been diagnosed (by a doctor/health professional) with asthma (an allergic

respiratory disease)?" The response was then categorised as having asthma (coded as 1) or no asthma (coded as 0).

### Explanatory variables

The factors considered in this study included age, ethnicity, marital status, sex, father's educational level, religion, health status report, household wall type, household source of drinking water, type of toilet facilities, household cooking fuel, toilet facilities shared, and household floor types.

Age was categorized into six age groups (18–29; 30–39; 40–49; 50–59; 60–69; ≥70), ethnicity into five main groupings (Akan, Ewe, Ga-Adangbe, Guan, Northern dialects), along with marital status (never married; currently married; co-habiting; separated/divorced; widowed), sex (male versus female), father's educational level (no formal education; primary or less; secondary or higher), religion (some religion versus no religion), health status report (moderate versus bad/very bad), difficulty with self-care (mild; moderate; severe/extreme), household wall type (durable material versus non-durable material), household source of drinking water (piped versus non-piped), type of toilet facilities (flush versus non-flush), household cooking fuel (wood–primitive fuel; coal/charcoal/kerosene–transition fuel; electricity/gas–advanced fuel), toilet facilities shared (shared versus not shared), and household floor types (hard floor versus earth floor). Consideration of these variables were informed by the literature on factors that might influence chronic disease outcomes like asthma, especially in lower income countries.

### Statistical analysis

Descriptive statistics were used to summarize the distribution of selected background characteristics of respondents. Categorical variables were summarised using frequencies with their associated percentages. Further analyses were conducted to examine individual and household-level factors that might be significantly associated with asthma and explored unobserved household level effects on the outcome. Single level and multilevel (mixed effects) binary logistic regression models were applied on 4621 individuals residing in 3970 households with complete measurements on asthma variable as well as complete measurements on potential explanatory variables considered in the final models. The extension of the single level binary logistic regression model to the multilevel logistic regression model is warranted because of the hierarchical structure of the SAGE dataset where we have individuals nested within households. Specifically, we applied random intercept multilevel logistic regression models to examine possible differences in asthma among individuals across households while simultaneously identifying potential risk factors. Thus, the multilevel modelling approach [35] placed particular emphasis on household level differences in the risk of asthma among individuals and the extent of nesting of asthma within a household which cannot be achieved through a single level logistic regression model.

The household-level Variance Partition Coefficient (VPC) [36] which measures the amount of variation in asthma among individuals from the random intercept multilevel logistic regression model is given by VPC = (household-level variance/ (household-level variance + individual-level variance)). Using the random intercept multilevel logistic regression, this quantity also coincides with the Intra-household Correlation Coefficient (ICC) which measures similarity in asthma episodes among individuals belonging to the same household. The individual-level residual is assumed to follow a standard logistic distribution with mean zero and variance $\pi^2/3$, where $\pi = 3.14$ [37].

Model parameters were obtained using maximum likelihood. Identity covariance structure provided a good fit to the data in the random intercept multilevel logistic model. The goodness of fit for the fitted models was examined using a likelihood ratio test (LRT), Akaike Information Criterion (AIC) and Bayesian Information Criterion (BIC). Variance inflation factor (VIF) was used to check multicollinearity, and a VIF value below 10 was considered acceptable [38]. All the analyses were performed using STATA Version 14 [39]. P-value <0.20 on a univariable logistic regression was considered to select candidate set of risk factors for multivariable logistic regression analysis. P-value < 0.05 was used to declare statistical significance.

## Ethical approval and consent to participate

SAGE was approved by the World Health Organization's Ethical Review Board (reference number RPC149) and the Ethical and Protocol Review Committee, College of Health Sciences, University of Ghana, Accra, Ghana. Written informed consent was obtained from all study respondents. All methods were performed in accordance with the relevant guidelines and regulations.

## Patients and public involvement statement

The questionnaire used for the SAGE Wave 2 was modified from that of SAGE Wave 1 due to patient experiences and priority lessons learnt. The design of SAGE Wave 2 was informed by the involvement of patients in Wave 1, modifications made were based on patient priorities. Recruitment of patients and conduct of the study was by the WHO SAGE Ghana Team. The WHO SAGE Ghana Team organizes national stakeholders meeting to disseminate the findings of the national survey. A report of the national survey based on all data collected is provided to the general public and available on the WHO SAGE website.

## Results

### Sample characteristics

Out of the 4670 individual respondents, 102 (2.2%) reported an asthma diagnosis. Among those aged 50 years or older, 85 (2.4%) had asthma and 1.5% among those aged 18–49 years. A majority (75.5%) of respondents were 50 years and older while 2745 (58.8%) of respondents were female. Among the participants residing in rural and urban communities, 53 (1.1%) and 49(1.1%) respectively had asthma. Majority (53.6%) of households use wood as cooking fuel. A total of 283 (6.1%) of the respondents ever smoked tobacco or used smokeless tobacco. Over 48% of the respondents were Akan, with the Guan ethnicity in the minority (4.1%). Fifty-six percent reported being currently married and 74% had fathers with no formal education. Over 97% reported following some form of religion. Household characteristics included durable material for walls (62.8%), non-flush toilets (84.4%), shared toilets (76.6%) and hard floor (86.4%) (Table 1).

### Univariable analyses

The results of univariable (unadjusted) analyses are presented in Table 2. Significant risk factors in the unadjusted model were age, ethnicity, religion, present health state, difficulty with self-care, source of drinking water and type of toilet facility in households. Those aged 60–69 (OR = 5.22, 95% CI: 1.24, 21.95) and 70 or more (OR = 5.56, 95% CI: 1.33, 23.26) years had increased odds of having asthma compared to those aged 18–29 years. Ga-Adangbe respondents (OR = 1.65, 95% CI: 1.01, 2.71) had higher odds of having asthma compared to their counterparts who are Akan while those with Northern dialect had lower odds of having asthma

**Table 1. Distribution (n, %) of selected background characteristics of respondents.**

| Characteristics | n(%) | Had asthma | No asthma |
|---|---|---|---|
| | | n (%) | n (%) |
| **All ages** | - | 102(2.18) | 4568(97.82) |
| **Aged ≥ 50 years** | 3575(75.50) | 85(2.41) | 3448(97.59) |
| **Aged 18–49 years** | 1160(24.50) | 17(1.50) | 1120(98.50) |
| **Place of residence** | | | |
| Rural | 3346(57.74) | 53(1.13) | 2695(57.71) |
| Urban | 2449(42.26) | 49(1.05) | 1873(40.11) |
| **Cooking fuel** | | | |
| Gas/electric | 642(11.91) | 14(0.30) | 540(11.69) |
| Coal/charcoal/kerosene | 1861(34.53) | 41(0.89) | 1558(33.72) |
| Wood | 2887(53.56) | 45(0.97) | 2423(52.43) |
| **Sex** | | | |
| Male | 1925(41.22) | 41(0.88) | 1884(40.34) |
| Female | 2745(58.78) | 61(1.31) | 2684(57.47) |
| **Ever smoked tobacco or used smokeless tobacco** | | | |
| Yes | 283(6.06) | 8(0.17) | 275(5.89) |
| No | 4388(93.94) | 94(2.01) | 4293(91.93) |
| **Ethnicity** | | | |
| Akan | 2265(48.50) | 55(1.18) | 2210(47.32) |
| Ewe | 272(5.82) | 4(0.09) | 268(5.74) |
| Ga-Adangbe | 583(12.48) | 23(0.49) | 560(11.99) |
| Guan | 192(4.11) | 1(0.02) | 191(4.09) |
| Northern dialect | 1358(29.08) | 19(0.41) | 1339(28.67) |
| **Marital status** | | | |
| Never married | 425(9.10) | 6(0.13) | 419(8.97) |
| Currently married | 2597(55.61) | 50(1.07) | 2547(54.54) |
| Cohabiting | 64(1.37) | 0(0) | 64(1.37) |
| Separated/divorced | 525(11.24) | 17(0.36) | 508(10.88) |
| Widowed | 1059(22.68) | 29(0.62) | 1030(22.06) |
| **Educational status** | | | |
| No formal education | 3459(74.07) | 81(1.73) | 3378(72.33) |
| Primary or less | 570(12.20) | 6(0.13) | 564(12.08) |
| Secondary or higher | 641(13.73) | 15(0.32) | 626(13.40) |
| **Religion** | | | |
| Some religion | 4541(97.24) | 93(1.99) | 4448(95.25) |
| No religion | 129(2.76) | 9(0.19) | 120(2.57) |
| **Wall type** | | | |
| Durable material | 2917(62.81) | 68(1.46) | 2849(61.35) |
| Non-durable material | 1727(37.19) | 33(0.71) | 1694(36.48) |
| **Source of drinking water** | | | |
| Piped | 2337(50.32) | 49(1.06) | 2288(49.27) |
| Non-piped | 2307(49.68) | 52(1.12) | 2255(48.56) |
| **Toilet facilities** | | | |
| Flush toilets | 722(15.57) | 23(0.50) | 699(15.08) |
| Non-flush toilets | 3914(84.43) | 78(1.68) | 3836(82.74) |
| **Shared toilet** | | | |
| Yes | 3003(76.55) | 67(1.71) | 2936(74.84) |

(*Continued*)

**Table 1.** (Continued)

| Characteristics | n(%) | Had asthma | No asthma |
|---|---|---|---|
| | | n (%) | n (%) |
| No | 920(23.45) | 24(0.61) | 896(22.84) |
| **Floor** | | | |
| Hard floor | 4006(86.39) | 90(1.94) | 3916(84.45) |
| Earth floor | 631(13.61) | 11(0.24) | 620(13.37) |

compared to those who are Akan. Those with no religion (OR = 3.59, 95% CI: 1.77, 7.28) had increased odds of having asthma compared to those with some religion. Respondents who rated themselves as having moderate (OR = 1.76, 95% CI: 1.13, 2.75) and bad/very bad (OR = 2.75, 95% CI: 1.58, 4.80) health state had increased odds of having asthma compared to those who rated themselves as good/very good. Those who rated their difficulty with self-care as severe/extreme (OR = 3.49, 95% CI: 1.23, 9.88) had increased odds of having asthma compared to those who rated themselves as none (i.e. no difficulty). Those residing in households with non-flush toilet facility (OR = 0.62, 95% CI: 0.39, 0.99) had lower odds of having asthma compared to those with flush toilets. We did not find marital status, sex, father's educational status, household wall type, cooking fuel, shared toilets and floor types significantly associated with asthma episodes.

## Multivariable analyses

In the multivariable (adjusted) analyses only age, father's education, religion and type of household toilet facilities were significantly associated with asthma episodes. The odds of having asthma was higher among those aged 60–69 (OR = 4.49, 95% CI: 1.03, 19.55) years compared to those aged 18–29 years. Having a father with primary education or less (OR = 0.40, 95% CI: 0.17, 0.94) decreased the odds of having asthma compared to those with no formal education. Having no religion (OR = 2.52, 95% CI: 1.18, 5.41) increased the odds of having asthma compared to those with some religion. Residing in households with non-flush toilet facility (OR = 0.58, 95% CI: 0.35, 0.96) lower the odds of having asthma compared to those with flush toilets (Table 2).

## Multilevel analyses

We extended the single level multivariable model to a multilevel model (Tables 2 and 3). This accounted for the clustering of individuals within the same households. Results from the multilevel showed that only age, father's education, religion and type of household toilet facilities were significantly associated with asthma. Though similar findings were observed in the models, their effect sizes were different. In the multilevel model, the odds of having asthma was higher among those aged 60–69 (OR = 5.06, 95% CI: 1.06, 24.07) years compared to those aged 18–29 years. Having a father with primary education or less (OR = 0.38, 95% CI: 0.15, 0.95) decreased the odds of having asthma compared to those with no formal education. Having no religion (OR = 2.93, 95% CI: 1.15, 7.47) increased the odds of having asthma compared to those with some religion. Residing in households with non-flush toilet facility (OR = 0.55, 95% CI: 0.31, 0.98) lower the odds of having asthma compared to those with flush toilets (Table 2).

Comparing the single level multivariable model to the multilevel model, we observed a p-value of 0.0477 which is below 0.05, suggesting that the multilevel model provided a good fit to the data. Thus, the multilevel model was preferred among the competing models suggesting strong household-level variation in asthma.

**Table 2. Odds ratio for asthma: Single-level and multilevel binary logistic regression models.**

| | Univariable model | | Multivariable model | | Multilevel model | |
|---|---|---|---|---|---|---|
| Characteristics | UOR (95% CI) | P-value | AOR (95% CI) | P-value | AOR (95% CI) | P-value |
| **Age** | | 0.073 | | | | |
| 18–29 | ref | | ref | | ref | |
| 30–39 | 4.01(0.83–19.44) | 0.085 | 3.91(0.77–19.8) | 0.1 | 4.32(0.77–24.15) | 0.096 |
| 40–49 | 3.47(0.73–16.42) | 0.117 | 3.37(0.7–16.32) | 0.131 | 3.72(0.7–19.77) | 0.123 |
| 50–59 | 3.1(0.72–13.27) | 0.128 | 2.93(0.66–12.96) | 0.156 | 3.19(0.67–15.25) | 0.147 |
| 60–69 | 5.22(1.24–21.95) | 0.024* | 4.49(1.03–19.55) | 0.045* | 5.06(1.06–24.07) | 0.042* |
| 70 or more | 5.56(1.33–23.26) | 0.019* | 3.83(0.87–16.98) | 0.077 | 4.32(0.89–20.93) | 0.069 |
| **Ethnicity** | | 0.005** | | | | |
| Akan | ref | | ref | | ref | |
| Ewe | 0.6(0.22–1.67) | 0.327 | 0.56(0.2–1.58) | 0.273 | 0.53(0.17–1.64) | 0.267 |
| Ga-Adangbe | 1.65(1.01–2.71) | 0.047* | 1.53(0.91–2.57) | 0.112 | 1.63(0.89–2.98) | 0.116 |
| Guan | 0.21(0.03–1.53) | 0.123 | 0.25(0.03–1.84) | 0.173 | 0.24(0.03–1.89) | 0.174 |
| Northern dialect | 0.57(0.34–0.96) | 0.036* | 0.67(0.39–1.16) | 0.151 | 0.65(0.35–1.19) | 0.162 |
| **Marital status** | | 0.177 | | | | |
| Never married | ref | | ref | | ref | |
| Currently married | 0.76(0.52–1.13) | 0.177 | 0.85(0.56–1.28) | 0.432 | 0.84(0.53–1.34) | 0.471 |
| **Sex** | | 0.832 | - | | - | |
| Male | ref | | - | | - | |
| Female | 1.04(0.7–1.56) | 0.832 | - | | - | |
| **Father's educational status** | | 0.158 | - | | - | |
| No formal education | ref | | ref | | ref | |
| Primary or less | 0.44(0.19–1.02) | 0.056 | 0.4(0.17–0.94) | 0.036* | 0.38(0.15–0.95) | 0.039* |
| Secondary or higher | 1(0.57–1.75) | 0.998 | 0.99(0.54–1.82) | 0.971 | 1.01(0.51–2) | 0.984 |
| **Religion** | | <0.001*** | | | | |
| Some religion | ref | | ref | | ref | |
| No religion | 3.59(1.77–7.28) | <0.001*** | 2.52(1.18–5.41) | 0.017* | 2.93(1.15–7.47) | 0.024* |
| **Health state** | | 0.001** | | | | |
| Good/very good | ref | | ref | | ref | |
| Moderate | 1.76(1.13–2.75) | 0.013* | 1.43(0.88–2.31) | 0.147 | 1.46(0.85–2.5) | 0.169 |
| Bad/very bad | 2.75(1.58–4.8) | <0.001*** | 1.93(0.99–3.76) | 0.052 | 1.97(0.92–4.21) | 0.079 |
| **Self-care** | | 0.049* | | | | |
| None | ref | | ref | | ref | |
| Mild | 0.67(0.35–1.3) | 0.234 | 0.6(0.3–1.23) | 0.165 | 0.56(0.26–1.23) | 0.148 |
| Moderate | 1.35(0.58–3.13) | 0.486 | 0.94(0.38–2.32) | 0.895 | 0.91(0.32–2.54) | 0.855 |
| Severe/extreme | 3.49(1.23–9.88) | 0.019* | 2.03(0.64–6.43) | 0.23 | 2.18(0.54–8.88) | 0.275 |
| **Wall** | | 0.343 | - | | - | |
| Durable material | ref | | - | | - | |
| Non-durable material | 0.82(0.54–1.24) | 0.343 | - | | - | |
| **Source of drinking water** | | 0.713 | - | | - | |
| Piped | ref | | - | | - | |
| Non-piped | 1.08(0.73–1.6) | 0.713 | - | | - | |
| **Toilet facilities** | | 0.046* | - | | - | |
| Flush toilets | ref | | ref | | ref | |
| Non-flush toilets | 0.62(0.39–0.99) | 0.046* | 0.58(0.35–0.96) | 0.033* | 0.55(0.31–0.98) | 0.041* |
| **Cooking fuel** | | 0.237 | | | | |
| Gas/electricity | ref | | - | | - | |

*(Continued)*

**Table 2.** (Continued)

| Characteristics | Univariable model | | Multivariable model | | Multilevel model | |
|---|---|---|---|---|---|---|
| | UOR (95% CI) | P-value | AOR (95% CI) | P-value | AOR (95% CI) | P-value |
| Coal/charcoal/kerosene | 1.02(0.55–1.88) | 0.962 | - | | - | |
| Wood | 0.72(0.39–1.31) | 0.281 | - | | - | |
| **Shared toilet** | | 0.506 | - | | - | |
| Yes | ref | | - | | - | |
| No | 1.17(0.73–1.88) | 0.506 | - | | - | |
| **Floor** | | 0.422 | - | | - | |
| Hard floor | ref | | - | | - | |
| Earth floor | 0.77(0.41–1.45) | 0.422 | - | | - | |

UOR: unadjusted odds ratio, AOR: adjusted odds ratio, CI: confidence interval, ref: reference category

*: p-value<0.05

**: p-value<0.01

***: p-value<0.001.

The variance partition coefficients (VPC) which coincides with the intra-class correlation coefficient (ICC) is presented in Table 3. The results showed that over 40% of variance in asthma episodes could be attributable to residual household-level variations after adjusting for individual and household level factors in our model.

## Discussion

The prevalence of asthma from a nationally representative study among those aged 18 years and older for the period under study (2014–2015) was 2.2%, with a prevalence of 2.4% among adults aged ≥50 years and 1.5% among those aged 18–49 years. The odds of having asthma was highest in the 60–69 year age group in the adjusted models. Protective factors were having a father with some form of education at the primary level or less and possession of a non-flush toilet.

The present study found the asthma prevalence rate among older adults was consistent with a multi-site study conducted earlier (2007–2010) [40], but lower than corresponding rates reported in independent surveys conducted in other African countries: 2.7% in Cameroon (≥19 years) [7], 4.3% in Gambia (≥15 years) [8], 5.3% in Botswana (10–64 years) [9], 6.5% in Tunisia (2–52 years) [10], 6.8% in Uganda (≥35 years) [11], 6.9% in Kinshasa (≥18 years) [12] and 15.2% in Nigeria (18–65 years) [13]. The asthma prevalence rate in the present study is also lower than has been reported in United States, 7.7% (≥ 18 years) [41] and 7.2% in Israel (≥21 years) [42], but it exceeds the asthma prevalence rate of 1.24% reported in China (≥14 years) [31]. The prevalence rates of 1.50%, 2.20% and 2.40%, respectively for those aged 18–49, ≥18 and ≥50 years reported in the present study was also lower than 3.65% (doctor-diagnosed asthma) and 3.77% (treated asthma) reported for Ghana in the World Health Survey, 2002 and 2003. The survey involved 70 of the 192 WHO member states, but participants were limited to adults aged 18-45years, which may contribute to the differences observed [43].

**Table 3. Variance component analysis from the multilevel binary logistic regression model.**

| Variance components | Estimate | 95% confidence interval | Intraclass correlation Coefficient (ICC) |
|---|---|---|---|
| Individual level (IL) | 3.29 | - | - |
| Household level (HL) | 2.22 | 0.55, 9.04 | HL/(HL+IL)*100 = 40.29% |

Notwithstanding the low prevalence rate estimated for asthma in the study population, the projected increase in the population of older adults underscores the need to ensure that exposure to modifiable risk factors are minimized.

These differences could be attributed to methodological differences (for example: definition of asthma and/or ascertainment of asthma cases by self-report, physician diagnosed or algorithm-based; whether or not a nationally representative sample was used) and differences in environmental exposures. For instance, an earlier multi-site study conducted in six low- and middle-income countries (China, Ghana, India, Mexico, Russia and South Africa) to determine the prevalence of NCDs found that for all six conditions (angina, arthritis, asthma, chronic lung disease, depression and hypertension), the algorithm/measured test prevalence was higher than self-reported prevalence of NCDs. Variations in asthma prevalence rates have been attributed to environmental exposures including microorganisms, pollutants, allergens and diet which influence the disease expression in susceptible persons [44].

Individuals in the age group 60–69 years demonstrated a five-fold increase in odds of having asthma episodes (OR = 5.06, 95% CI: 1.06, 24.07) compared to individuals in the youngest age group, 18–29 years. Furthermore, a study conducted in China among adults ≥15 years, the highest prevalence (114/2809; 4.06%) was reported among adults aged ≥60 years [23]. A multi-centre study in India demonstrated increased odds of having asthma with increasing age [45], however the present study showed a decline in odds from 18–59 years and increased in the age group 60–69 years, reaching statistical significance. This finding is further supported by a study in North Africa which reported that asthma prevalence rates were highest at extremes of age. Peak prevalence rates were reported in children below the age of 16 years and in adults >54 years in Algeria and among adults aged >54 years in Tunisia [14].

The relationship between religion and asthma is uncertain. Rather it has been used to explain self-management behaviours, medication adherence and has been perceived as a coping strategy in asthma where studies have demonstrated either a potentially positive influence [46, 47] or a negative effect [48]. The positive influence of religion on individual health appears to be mediated through provision of social ties and support, access to the social capital of religious groups, enhancing health promoting behaviours and positive emotions [49–51]. Consequently, the absence of religious affiliation or involvement may foster an environment that impacts negatively on health (for example, smoking and lack of physical activity) particularly in older adults. This might predispose to the expression of asthmatic phenotypes, where there is genetic disposition, atopy and exposure to environmental triggers. The nearly three-fold increase in odds of having asthma calls for further investigation to explain the mechanism underlying this finding.

Having a father with primary education or less as the highest level of education (compared to no formal education) had a protective effect in the present study. The influence of paternal education on asthma in adults is unclear. An educated parent is more aware of and better able to understand the disease, recognize its symptoms and support its daily management [52]. This would include among others, access to appropriate healthcare. Education further motivates the avoidance of health risks [53] such as exposure to environmental agents that trigger attacks. Education also erodes traditional beliefs regarding the causes of diseases as being due to spiritual forces, their agents or ancestors which often results in the patronage of prayer camps or alternative providers believed to have a cure for all kinds of diseases [54]. Lack of formal education is often closely linked to a low socioeconomic status which predisposes individuals to adverse conditions often injurious to health. This is particularly important in older adults as the biological process of aging has been associated with increased vulnerability to sickness and limited mobility [55].

Individuals reporting the use of non-flush toilets in households were 56% less likely to have asthma episodes [OR = 0.44, 95%CI 0.31, 0.98] compared to flush toilets and therefore this was

a protective factor. Flush toilets tend to produce aerosols in substantial quantities when flushed. Some of these aerosols, containing microbes, desiccate to become droplet nuclei above the toilet within 5 minutes and up to 90 minutes post-flush, with the toilet seat raised [56]. They are carried in air currents as viable forms small enough to get translocated through the nasal passages during inhalation. Exposure of respiratory epithelia to microbes stimulate the antigen presenting cells to induce an immune response consistent with inflammation characterized by increased eosinophils, basophils, and T helper 2 cells. Furthermore, breaches in epithelial integrity and inability of the macrophages to clear microbes in the airways may propagate inflammatory responses [57]. These protective mechanisms in the lungs may decline with age. Non-flush toilets are less likely to generate aerosols, unlike flush toilets that generate aerosols per flush. The ability to cause infection depends on the ability of the pathogen to survive environmental conditions, the number of organisms inhaled, their virulence and the immune status of the host. Available evidence suggests the potential for airborne transmission of norovirus, Severe Acute Respiratory Syndrome (SARS) coronavirus and influenza, since these can be shed in faeces and vomit which is often disposed of in toilets [56]. Infections of the respiratory tract such as influenza can trigger asthmatic episodes in predisposed persons. The toilet is one of the areas in the built environment that contributes to the microbiome to which a susceptible adult can be exposed. Ethnicity, present health state, difficulty with self-care and source of drinking water did not achieve statistical significance in the multivariable and multilevel analysis.

The results make a strong case for ensuring universal access to education up to at least a primary level as currently implemented is sustained to inform healthy choices and ensure progress in meeting Sustainable Development Goals. Ensuring that flush toilets are either flushed with seats closed or non-flush options are available options to households with older adults should be driven by the Ministry of Health through advocacy and implemented at district level. Intersectoral collaboration with the Ministry of Water and Sanitation, Ministry of Local Government and Rural Development, Environmental Protection Agency, Ministry of Finance and relevant development partners will be required for implementation. Religious affiliation and involvement in religious activity should be promoted to harness potential benefits for support to older adults. Religious bodies can extend outreach services to older adults through home visits either independently or as an integrated service with healthcare facilities.

The strengths of the present study include: the use of a nationally representative sample selected by probability based multi-stage sampling which allows generalization of the results. Furthermore, a standardized questionnaire used in the study also permits the comparability of results across countries in the study. The use of multilevel modelling approach controlled for the effect of clustering in households, resulting in an unbiased standard error of the estimates which will help avoid spurious significant risk factors considered in the model. Also, combining individual and household-level data allows identification of both individual and household level factors that might influence the outcome. The limitations in the study include the use of self-report, however the discomforting nature of the symptoms make it unlikely to be missed by the affected individual and a combination of other methods of ascertainment might have yielded additional cases. The nature of our data and the modelling approach did not allow for drawing causation effects. The use of $p < 0.2$ as the stopping rule to identify candidate set of covariates in the bivariate model to be included in the multivariable model might not provide optimal variable selection for the covariates, although previous studies have provided a strong recommendation for using p-values in the range of 0.15–0.20 [58] as used in our study. One major challenge in asthma epidemiologic studies in developing countries is the extent of asthma control and proper asthma management including asthma controller treatment availability, compliance to treatment, comorbidities, and impact on quality of life. A previous study

in Ghana has reported daily asthma symptom occurrence of 11.7%, reliever medication rates were estimated as 18.2%, with regular use at 56.5% among the 77 asthmatic patients involved [59].

## Conclusion

Risk factors associated with asthma episodes among adults in Ghana were being aged within 60–69 years and absence of religious affiliation. Level of father's education with primary or less as the highest level and having a non-flush toilet were protective factors. Ethnicity, present health state, difficulty with self-care and source of drinking water were not significantly independently related to asthmatic episodes in this study.

Overall, individual as well as household factors were seen to influence the prevalence of asthma in this national survey. Clinical management of these patients in health facilities should consider household factors in addition to individual level factors. In addition, national efforts in promoting dry sanitation systems and appropriate use of flush systems to minimize aerosol dispersion for the citizenry is worth considering.

## Acknowledgments

We are grateful to the World Health Organization and her partners who made the SAGE study possible and provided us with the data at no cost. We are also thankful to all respondents and interviewers who participated in the SAGE survey in Ghana. The Ministry of Health, Ghana, is supportive of SAGE. The University of Ghana's Department of Community Health contributed training facilities, data entry support, and storage of materials. The Ghana Statistical Service provided the sampling information for the sampling frame and updates.

## Author Contributions

**Conceptualization:** Justice Moses K. Aheto, Emilia A. Udofia.

**Data curation:** Justice Moses K. Aheto.

**Formal analysis:** Justice Moses K. Aheto.

**Investigation:** Justice Moses K. Aheto, Emilia A. Udofia, Alfred E. Yawson.

**Methodology:** Justice Moses K. Aheto.

**Resources:** Justice Moses K. Aheto, Emilia A. Udofia, Alfred E. Yawson.

**Software:** Justice Moses K. Aheto.

**Validation:** Justice Moses K. Aheto.

**Writing – original draft:** Justice Moses K. Aheto, Emilia A. Udofia.

**Writing – review & editing:** Justice Moses K. Aheto, Eugene Kallson, George Mensah, Minicuci Nadia, Naidoo Nirmala, Somnath Chatterji, Paul Kowal, Richard Biritwum, Alfred E. Yawson.

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
