## [Decision Letter · Decision Letter 0]

6 Aug 2020

PONE-D-20-10825

Prevalence, socio-demographic and environmental determinants of asthma in 4621 Ghanaian adults: Evidence from Wave 2 of the World Health Organization’s Study on global AGEing and adult health.

PLOS ONE

Dear Dr. Aheto,

Thank you for submitting your manuscript to PLOS ONE. After careful consideration, we feel that it has merit but does not fully meet PLOS ONE’s publication criteria as it currently stands. Therefore, we invite you to submit a revised version of the manuscript that addresses the points raised during the review process.

Please consider whether you are able and willing to incorporate the comment of Reviewer 2 on using wave 1 in addition for avoing the limitations of a cross-sectional study. The decision is up to you, but please comment on this issue why you decided to exclude ot take wave 1 into account.

We look forward to receiving your revised manuscript.

Kind regards,

Florian Fischer

Academic Editor

PLOS ONE

Journal Requirements:

"Financial support was provided by the US National Institute on Aging through

Interagency Agreements (OGHA 04034785; YA1323-08-CN-0020; Y1-AG-1005-01) with the

World Health Organization and a Research Project Grant (R01 AG034479- 64401A1). WHO

contributed financial and human resources to SAGE. The Ministry of Health, Ghana, is

supportive of SAGE. The University of Ghana’s Department of Community Health contributed

training facilities, data entry support, and storage of materials. The Ghana Statistical Office

provided the sampling information for the sampling frame and updates."

"The authors received no specific funding for this work"

We note that one or more of the authors are employed by a commercial company: 3Deloitte Consulting, West Africa Deloitte & Touche.

3.1. Please provide an amended Funding Statement declaring this commercial affiliation, as well as a statement regarding the Role of Funders in your study. If the funding organization did not play a role in the study design, data collection and analysis, decision to publish, or preparation of the manuscript and only provided financial support in the form of authors' salaries and/or research materials, please review your statements relating to the author contributions, and ensure you have specifically and accurately indicated the role(s) that these authors had in your study. You can update author roles in the Author Contributions section of the online submission form.

3.2. Please also provide an updated Competing Interests Statement declaring this commercial affiliation along with any other relevant declarations relating to employment, consultancy, patents, products in development, or marketed products, etc. 

Reviewers' comments:

Reviewer's Responses to Questions

**Comments to the Author**

1. Is the manuscript technically sound, and do the data support the conclusions?

Reviewer #1: Yes

Reviewer #2: Yes

2. Has the statistical analysis been performed appropriately and rigorously? 

Reviewer #1: Yes

Reviewer #2: No

3. Have the authors made all data underlying the findings in their manuscript fully available?

Reviewer #1: Yes

Reviewer #2: Yes

4. Is the manuscript presented in an intelligible fashion and written in standard English?

Reviewer #1: Yes

Reviewer #2: Yes

5. Review Comments to the Author

Reviewer #1: I felt this was a well done study. The manuscript provides helpful information on Ghanaian health that contains actionable areas for future public health interventions. The addition of a) a map of Ghana, and b) more geographical context for from where within Ghana respondents live would strengthen the paper.

Reviewer #2: This is an exploratory study examining the prevalence, socio-demographic and environmental determinants of asthma in Ghana. My main concerns include 1) the cross-sectional design, 2) the lack of considerations of neighborhood-level risk factors, and 3) the variable selection procedure used. Below please find my detailed comments.

1. Page 3, background, reference 1: please update the reference and cite the most recent statistics from the global burden of disease study.

2. Page 5, study population: please briefly describe the sampling strategy used in SAGE.

3. Page 5, study population and outcome variable: since SAGE is a longitudinal study, it might be better to use data from both wave 1 and wave 2 to identify incident asthma cases. The current analsyes depending on the asthma question only from the wave 2 made this a cross-sectional study, which substantially limited the significance of this study.

4. Page 5, explanatory variables: as this is an exploratory study, it is unclear why the authors only included these variables. Why neighborhood-level factors were not included? Many established risk factors mentioned in the background section were not included.

5. Page 7, statistical analyses: the authors only included variables with a p-value<0.2 from the univariable regression model in the multivariable regression model. This variable selection approach is problematic. More advanced approach such as the elastic net model or Lasso model with hyperparameters determined by cross-validations should be used to perform the variable selection. e.g., the glmmLasso package in R can be used to implement Lasso for gelearized linear mixed models.

6. Page 15: one major limitation is the cross-sectional design of this study: the current exposures to the risk factors may not be representative of the historical exposures before the onset of asthma. This needs to be discussed in the limitation section. Again, if asthma data is available from wave 1, I highly recommend the authors to use the data to identify incident asthma cases to avoid the cross-sectional design of the study.

6. PLOS authors have the option to publish the peer review history of their article (what does this mean?). If published, this will include your full peer review and any attached files.

Reviewer #1: No

Reviewer #2: No

---

## [Author Response · Author response to Decision Letter 0]

10 Sep 2020

Responses from the authors:

First, we extend our profound gratitude and thanks to the Editors and the Reviewers for their efforts and valuable comments on our paper. 

We have given your comments the needed attention and action and are happy to resubmit the revised manuscript for your kind consideration and subsequent publication in your cherished journal, and together we can all help in addressing this serious public health challenge facing millions of people in most developing countries like Ghana. 

Below are our responses to the comments in italics and you will also find the needed changes in the main report tracked in colour using the tracking facility in word. In addition, we have made few other changes to the paper with the sole aim of enhancing the understanding of our readers and these were also tracked in colour. 

Journal Requirements:

 Response: we have revised the manuscript to address this (e.g. see the tittle page, abstract and the introduction, materials and methods). 

"Financial support was provided by the US National Institute on Aging through

Interagency Agreements (OGHA 04034785; YA1323-08-CN-0020; Y1-AG-1005-01) with the

World Health Organization and a Research Project Grant (R01 AG034479- 64401A1). WHO

contributed financial and human resources to SAGE. The Ministry of Health, Ghana, is

supportive of SAGE. The University of Ghana’s Department of Community Health contributed

training facilities, data entry support, and storage of materials. The Ghana Statistical Office

provided the sampling information for the sampling frame and updates."

"The authors received no specific funding for this work"

 Response: Thank you. The present study (the work in this manuscript) did not receive any funding. We were only acknowledging those who funded the original survey that generated the data which is serving as a secondary data for the present study. However, some of my co-authors were part of investigators during the main survey. We revised the manuscript by removing the funding statement under the acknowledgement section and also removed the ‘funding section’ from the manuscript (see lines 7-12, page 20). 

The acknowledgement now reads “We are grateful to the World Health Organization and her partners who made the SAGE study possible and provided us with the data at no cost. We are also thankful to all respondents and interviewers who participated in the SAGE survey in Ghana. The Ministry of Health, Ghana, is supportive of SAGE. The University of Ghana’s Department of Community Health contributed training facilities, data entry support, and storage of materials. The Ghana Statistical Service provided the sampling information for the sampling frame and updates.”

For the online funding statement, the following can be used or you can modify it to suit the journal requirement: “Funding was obtained for the main WHO SAGE Wave 2 Survey in Ghana with financial support provided by the US National Institute on Aging through Interagency Agreements (OGHA 04034785; YA1323-08-CN-0020; Y1-AG-1005-01) with the World Health Organization and a Research Project Grant (R01 AG034479- 64401A1). The funder provided support in the form of salaries for authors [RB, AEY], but did not have any additional role in the study design, data collection and analysis, decision to publish, or preparation of the manuscript. The specific roles of these authors are articulated in the ‘author contributions’ section. However, the work in this study did not receive any funding support. Also, Deloitte Consulting, West Africa Deloitte & Touche did not provide any financial support for the study and played no role in the study”. See response to point 3 below:

We note that one or more of the authors are employed by a commercial company: 3Deloitte Consulting, West Africa Deloitte & Touche.

Response: Thank you. Deloitte Consulting, West Africa Deloitte & Touche did not provide any financial support for the study and played no role in the study. Our co-author, Mr Kallson participated in this study in his capacity as an individual. Thus, he was not representing his company. We addressed this in our response to point 2 above. 

3.1. Please provide an amended Funding Statement declaring this commercial affiliation, as well as a statement regarding the Role of Funders in your study. If the funding organization did not play a role in the study design, data collection and analysis, decision to publish, or preparation of the manuscript and only provided financial support in the form of authors' salaries and/or research materials, please review your statements relating to the author contributions, and ensure you have specifically and accurately indicated the role(s) that these authors had in your study. You can update author roles in the Author Contributions section of the online submission form.

Response: Thank you. The commercial affiliation did not play any role in the study. We have updated the funding (We addressed this in our response to point 2 above) and the author contribution statements (lines 13-16, page 19) to reflect the changes required. 

We provided the following: “All authors except Mr Kallson declare that they have no conflict of interest. Though Mr Kallson was working with Deloitte Consulting, West Africa Deloitte & Touche at the time of the study, Deloitte Consulting did not provide any financial support for the study and did not play any role in the study design, data collection and analysis, preparation of the manuscript, or decision to publish. Mr Kallson participated in this study in his capacity as an individual and was not acting on behalf of Deloitte Consulting, and this does not alter our adherence to PLOS ONE policies on sharing data and materials.”

3.2. Please also provide an updated Competing Interests Statement declaring this commercial affiliation along with any other relevant declarations relating to employment, consultancy, patents, products in development, or marketed products, etc. 

 Response: Thank you. We have provided the updated funding and competing interests statement in the cover letter as requested. 

Response: There is no legal restrictions on the data known to the authors. The data is provided to researchers freely but the WHO SAGE Team only releases the data to researchers upon request directly made to WHO, and individual researchers granted permission to use the data are not allowed to make the data (in any form) available to third parties. Any third party interested in the data must apply directly to the WHO SAGE Team. We have included this in the cover letter as requested, and also in the manuscript. 

Under data availability, we inserted “Data is freely available upon making official request to WHO SAGE Team through the WHO website at http://www.who.int/healthinfo/sage/cohorts/en/. The data is provided to researchers freely but the WHO SAGE Team only releases the data to researchers upon request made directly to WHO, and individual researchers granted permission to use the data are not allowed to make the data (in any form) available to third parties. Any third party interested in the data must apply directly to the WHO SAGE Team”. See lines 1-2, page 19 and lines 1-4, page 20.

Response: Please, refer to our response to point 4(a) and also to the cover letter. 

Reviewers' comments:

Reviewer #1: I felt this was a well done study. The manuscript provides helpful information on Ghanaian health that contains actionable areas for future public health interventions. The addition of a) a map of Ghana, and b) more geographical context for from where within Ghana respondents live would strengthen the paper.

Response: Thank you for the useful comment. Unfortunately, the survey did not collect data on spatial coordinates (latitude and longitude) of the participants in the 2014/2015 study. Our current study is based on this secondary data and it is therefore not possible to show the locations of these participants on the map of Ghana in the present study. 

Reviewer #2: This is an exploratory study examining the prevalence, socio-demographic and environmental determinants of asthma in Ghana. My main concerns include 1) the cross-sectional design, 2) the lack of considerations of neighborhood-level risk factors, and 3) the variable selection procedure used. Below please find my detailed comments.

1. Page 3, background, reference 1: please update the reference and cite the most recent statistics from the global burden of disease study.

Response: Thank you. Revised as suggested. In the 2020 GBD study report, there is no combine prevalence and number of deaths for asthma and chronic obstructive pulmonary disease. As a result, we revised the text and provide the updated statistics from the 2020 report while maintaining the 2017 statistics also and both referenced. See lines 2-8 at page 3.

2. Page 5, study population: please briefly describe the sampling strategy used in SAGE.

Response: Thanks for the useful feedback. We revised the manuscript to reflect this. However, we effected this under data source and sampling because we believe is more appropriate under the data source compared to the study population. See line 14-20 at page 5. 

3. Page 5, study population and outcome variable: since SAGE is a longitudinal study, it might be better to use data from both wave 1 and wave 2 to identify incident asthma cases. The current analsyes depending on the asthma question only from the wave 2 made this a cross-sectional study, which substantially limited the significance of this study.

Response: The focus of this paper is to quantify unobserved household-level residual variations in asthma episodes while simultaneously identifying critical risk factors associated with the disease. We agreed that using data from wave 1 and 2 might provide additional information BUT there was a challenge where over 50% of those who participated in wave 1 conducted in 2007-2008 dropped out of the study in wave 2 conducted in 2014-2015 and new samples were drawn to replace some of them. Also, the secondary data we used did not uniquely identify participants in wave 1 only, wave 2 only and those in both wave 1 and 2. This introduced some analytical challenges because in analysing both wave 1 and wave 2, the modelling become much complex because we need to take into account that majority of those in wave 1 did not participate in wave 2 and that new samples were introduced in wave 2. Thus, there is a mixture of repeated cross-sectional and longitudinal designs which must be considered when modelling based on wave 1 and 2 but this will require availability of further data (e.g. variables that uniquely identify participants in wave 1 only, wave 2 only and those in both wave 1 and 2) and also this kind of analysis will require much more efforts than that of the present study. We will continue to engage the owners of the data and if successful, we will implement a modelling strategy that will account for the mixture of repeated cross-sectional and longitudinal designs nature of the data in our future studies. 

4. Page 5, explanatory variables: as this is an exploratory study, it is unclear why the authors only included these variables. Why neighborhood-level factors were not included? Many established risk factors mentioned in the background section were not included.

Response: We used secondary data in our analysis and except place of residence (urban/rural), no other neighbourhood/environmental covariates were available in the data. If the data were to contain geographical coordinates (latitudes/longitudes) of the respondents (this was not collected in the study), we could use that to extract neighbourhood/environmental/spatial covariates and include same in our model. Also, we explored all potential covariates in the available data through a combination of expert opinion, available literature, and statistical methodology for variable selection. 

5. Page 7, statistical analyses: the authors only included variables with a p-value<0.2 from the univariable regression model in the multivariable regression model. This variable selection approach is problematic. More advanced approach such as the elastic net model or Lasso model with hyperparameters determined by cross-validations should be used to perform the variable selection. e.g., the glmmLasso package in R can be used to implement Lasso for gelearized linear mixed models.

Response: Thank you for this suggestion. First of all, we used expert opinion and available literature on deciding potential predictors of asthma episodes to arrive at all the variables presented in Table 1. After that, we applied the statistical methodology for variable selection using p<0.2 in the univariable (bivariate) analysis as a threshold to identify candidate set of variables that will enter the multivariable model which is a standard approach. We will consider the elastic net model or Lasso model suggestion in our future research. 

6. Page 15: one major limitation is the cross-sectional design of this study: the current exposures to the risk factors may not be representative of the historical exposures before the onset of asthma. This needs to be discussed in the limitation section. Again, if asthma data is available from wave 1, I highly recommend the authors to use the data to identify incident asthma cases to avoid the cross-sectional design of the study.

Response: We already stated in the limitation that the nature of the data and the modelling approach could not allow for drawing effect of causation (see lines 19-20 at page 18), and we did not attempt to make any statement to suggest causation effect between the covariates and asthma episodes. We already provided response to the use of wave 1 and 2 in comment 3 which also addressed this.

---

## [Decision Letter · Decision Letter 1]

19 Nov 2020

PONE-D-20-10825R1

Prevalence, socio-demographic and environmental determinants of asthma in 4621 Ghanaian adults: Evidence from Wave 2 of the World Health Organization’s Study on global AGEing and adult health.

PLOS ONE

Dear Dr. Aheto,

Thank you for submitting your manuscript to PLOS ONE. After careful consideration, we feel that it has merit but does not fully meet PLOS ONE’s publication criteria as it currently stands. Therefore, we invite you to submit a revised version of the manuscript that addresses the points raised during the review process.

PLease take into cnsideration thje point raised by Reviewer 2.

We look forward to receiving your revised manuscript.

Kind regards,

Florian Fischer

Academic Editor

PLOS ONE

Reviewers' comments:

Reviewer's Responses to Questions

**Comments to the Author**

1. If the authors have adequately addressed your comments raised in a previous round of review and you feel that this manuscript is now acceptable for publication, you may indicate that here to bypass the “Comments to the Author” section, enter your conflict of interest statement in the “Confidential to Editor” section, and submit your "Accept" recommendation.

Reviewer #1: All comments have been addressed

Reviewer #2: (No Response)

2. Is the manuscript technically sound, and do the data support the conclusions?

Reviewer #1: Yes

Reviewer #2: No

3. Has the statistical analysis been performed appropriately and rigorously? 

Reviewer #1: Yes

Reviewer #2: No

4. Have the authors made all data underlying the findings in their manuscript fully available?

Reviewer #1: Yes

Reviewer #2: Yes

5. Is the manuscript presented in an intelligible fashion and written in standard English?

Reviewer #1: Yes

Reviewer #2: Yes

6. Review Comments to the Author

Reviewer #1: The manuscript has been improved and strengthened. I have rarely seen data on asthma in an African country and this work will add to that literature.

Reviewer #2: The authors did not adequately address my concerns, especially for the variable selection procedure. The response letter includes a statement that "we applied the statistical methodology for variable selection using p<0.2 in the univariable (bivariate) analysis as a threshold to identify candidate set of variables that will enter the multivariable model which is a standard approach", which is not true.

7. PLOS authors have the option to publish the peer review history of their article (what does this mean?). If published, this will include your full peer review and any attached files.

Reviewer #1: No

Reviewer #2: No

---

## [Author Response · Author response to Decision Letter 1]

23 Nov 2020

Review Comments to the Author

Reviewer #1: The manuscript has been improved and strengthened. I have rarely seen data on asthma in an African country and this work will add to that literature.

Response: Thank you.

Reviewer #2: The authors did not adequately address my concerns, especially for the variable selection procedure. The response letter includes a statement that "we applied the statistical methodology for variable selection using p<0.2 in the univariable (bivariate) analysis as a threshold to identify candidate set of variables that will enter the multivariable model which is a standard approach", which is not true.

Response: Thank you for pointing this out. Our decision to use threshold of a p-value <0.2 is based on the discussion in the literature on traditional stopping rule and suggested optimal p-values. The literature on this topic provided a strong recommendation for using a p-value in the range of 0.15–0.20 (Hosmer & Lemeshow, 2013), although using a higher significance level has the disadvantages that some unimportant variables may be included in the model (Hosmer & Lemeshow, 2013). Typically, the stopping rule for the traditional choice for significance level is considered to be between 0.05 and 0.10 though it was also established that the optimum value of the significance level to decide which variable to include in the multiple regression model is suggested to be p<1 (Chowdhury & Turin, 2020; Harrell, 2001), which actually exceeds the traditional choices. 

The above discussion in the literature is what informed our choice and that is what the authors meant by the statement "we applied the statistical methodology for variable selection using p<0.2 in the univariable (bivariate) analysis as a threshold to identify candidate set of variables that will enter the multivariable model which is a standard approach". Our choice of p-value<0.2 also lies between the maximum for the traditional rule and that of suggested p-value<1 which is an acceptable approach. 

After extensive reading, we observed that there are other techniques such as Lasso, Elastic net, and Ridge regression approaches based on machine learning techniques which could provide optimal variable selection which we did not do in our current study. However, this machine learning approaches of variable selection requires extensive learning of the procedure and the software for implementation, and the authors will surely explore this in their future research work. 

We revised the manuscript to highlight this under the limitations in lines 400-403 at page 18. Specifically, we inserted the statement “The use of p<0.2 as the stopping rule to identify candidate set of covariates in the bivariate model to be included in the multivariable model might not provide optimal variable selection for the covariates, although previous studies have provided a strong recommendation for using p-values in the range of 0.15-0.20 [58] as used in our study”.

References 

Hosmer DW, Lemeshow S, Sturdivant RX. Applied logistic regression. New York: John Wiley & Sons, Incorporated, 2013.

Chowdhury MZI, Turin TC. Variable selection strategies and its importance in clinical prediction modelling. Fam Med Community Health. 2020 Feb 16;8(1):e000262. doi: 10.1136/fmch-2019-000262. PMID: 32148735; PMCID: PMC7032893.

Harrell FE. Regression modeling strategies with applications to linear models, logistic regression, and survival analysis. New York: Springer, 2001.

---

## [Editor Report · Decision Letter 2]

25 Nov 2020

Prevalence, socio-demographic and environmental determinants of asthma in 4621 Ghanaian adults: Evidence from Wave 2 of the World Health Organization’s Study on global AGEing and adult health.

PONE-D-20-10825R2

Dear Dr. Aheto,

We’re pleased to inform you that your manuscript has been judged scientifically suitable for publication and will be formally accepted for publication once it meets all outstanding technical requirements.

Kind regards,

Florian Fischer

Academic Editor

PLOS ONE
---

## [Editor Report · Acceptance letter]

1 Dec 2020

PONE-D-20-10825R2 

Prevalence, socio-demographic and environmental determinants of asthma in 4621 Ghanaian adults: Evidence from Wave 2 of the World Health Organization’s Study on global AGEing and adult health. 

Dear Dr. Aheto:

I'm pleased to inform you that your manuscript has been deemed suitable for publication in PLOS ONE. Congratulations! Your manuscript is now with our production department. 

Kind regards, 

on behalf of

Dr. Florian Fischer 

Academic Editor

PLOS ONE